# An Innovative Approach to Determining the Contribution of Saharan Dust to Pollution

**DOI:** 10.3390/ijerph18116100

**Published:** 2021-06-05

**Authors:** Nicoletta Lotrecchiano, Vincenzo Capozzi, Daniele Sofia

**Affiliations:** 1DIIN-Department of Industrial Engineering, University of Salerno, Via Giovanni Paolo II, 132, 84084 Fisciano, Italy; nlotrecchiano@unisa.it; 2Department of Science and Technology, University of Naples “Parthenope”, Centro Direzionale di Napoli, Isola C4, 80134 Naples, Italy; vincenzo.capozzi@uniparthenope.it; 3Research Department, Sense Square Srl, 84084 Salerno, Italy

**Keywords:** air quality, Saharan dust, environmental pollution, mineral dust

## Abstract

Air quality is one of the hot topics of today, and many people are interested in it due to the harmful effects that environmental pollution has on human health. For this reason, in recent years, measurement systems based on advanced technology have been implemented to integrate national air quality networks. This study aimed to analyze the air quality data of the monitoring network of the regional agency for environmental protection of the Campania region (Italy), integrated with a monitoring station based on IoT technology to highlight criticalities in the levels of pollution. The data used was from the month of February 2021 and measured in a medium-large city in southern Italy. In-depth analyses showed that two events related to Saharan dust occurred, which led to an increase in the measured PM10 values.

## 1. Introduction

Europe’s industrialized regions are continuous sources of anthropogenic particulate matter. In addition to the gaseous emissions, these sources also produce fine powders. Particles can be inhaled while breathing, therefore high concentrations of atmospheric particles can have dangerous effects on human health [1]. However, not all particles are harmful, only those with small grain sizes.

Particulate material (PM) is a mixture of different components that vary both locally and regionally. PM can have various origins; it can be generated by natural phenomena, such as soil erosion or, more commonly, from vehicle combustion or industrial plant emissions. PM10 and PM2.5 are the two parts into which particulate matter is divided in particles with an aerodynamic diameter of ≤10 μm and ≤2.5 μm, respectively. Particles with a diameter less than 10 μm constitute the inhalable fraction, able to reach the broncho-tracheal area, while particles with a diameter less than 2.5 μm, which constitute the breathable fraction, can reach the alveoli lungs, conveying the substances of which they are composed into the body. PM10 is partly of the primary type, being entered directly into the atmosphere, and partly of the secondary type, being produced by physico-chemical transformations involving various substances such as SO_x_, NO_x_, VOC, and NH_3_, which determine its production and/or removal.

Particulate matter of anthropogenic origin largely belongs to the fraction with a diameter less than 2.5 μm, while desert clouds of dust contain a significant fraction with larger particle size; a considerable decrease in the PM2.5/PM10 ratio is, therefore, a contributing index of natural particulate matter of desert origin [2]. Saharan episodes are characterized by a large increase in the PM10 concentration, not followed by that of PM2.5 and PM1, with a consequent decrease in the PM2.5/PM10 ratio, an increase of the crustal elements (Al, Si, Ca, Ti, Fe) in the PM10 fraction, and an increase in the Si/Fe, Al/Fe, and Ti/Fe ratios [3,4]. Moreover, the effects of these Saharan events have repercussions, not only for outdoor air quality, but also for indoor air quality, which is affected by the high concentrations of aero-dispersed dust [5].

From a climatic point of view, the Mediterranean atmosphere (MED), as defined by Jeftic et al. [6], is characterized by rainy winter seasons and hot and dry summer seasons, which affect the continental areas that surround it. It is therefore full of particulates of anthropogenic origin, coming from the vast industrialized European regions, and of natural particulates of crustal origin, coming from the extensive arid and semi-arid areas of Africa and the Middle East, and of aerosols of marine origin, generated by the surface of the MED, and of volcanic dust, emitted by the main volcanoes of the basin, such as Etna and Stromboli [7]. From the studies carried out on the influence of Saharan dust on a large scale, it is highlighted that latitude is a very important factor that must be taken into account in studies concerning PM10 concentrations. This same difference can be seen between the number of Saharan transport events that take place in Spain and those that take place in Italy or England [8]. The possibility of there occurring an interference of the powder transport in the PM10 concentration is higher in Spain than in northern Italy, and much greater than in England. Various studies have involved Italy in recent years, highlighting how, depending on the areas investigated, the days of transport of dust increase from north to south [2]. Therefore, a need is highlighted, concerning European countries located at higher latitudes, to look for these signals, both in PM10 and PM2.5. The problem of dust can be analyzed using integrated approaches to define the link between the natural and the urban dust within the environment [9].

From this perspective, air quality monitoring systems have been introduced. These systems have evolved over the years using increasingly innovative technologies based on the principles of the Internet of things (IoT). The new smart monitoring devices [10] integrate the traditional monitoring stations implemented by the institution to create a monitoring network with high spatial and temporal resolution. The new monitoring networks, given their characteristics and small dimensions, allow their installation in many points of the urban or extra-urban context, optimizing their position [11]. The IoT-based sensors include a data protection system based on the blockchain principle that ensures data integrity [12]. Recently, the technology has developed to the point of making the monitoring stations dynamic, i.e., housed on a moving support vehicle that allows on-the-road real-time monitoring [13]. This new type of sensor can also be portable and worn by users so that they can directly assess their exposure to indoor and outdoor pollution [14]. Knowledge of airborne pollutant concentrations is a fundamental part of the definition of strategies for pollution reduction [15,16]. The pollutants being analyzed are commonly the average particulate PM10 and PM2.5, to which PM1 is sometimes added, as well as gases such as NO_2_, SO_2_, CO, H_2_S, O_3_, and VOC (volatile organic compounds). The measured air quality data, albeit in large quantities, are discontinuous in the territory and need to be integrated into modeling systems [17] that allow their analysis, forecasting, and spatialization. In particular, the in-depth analysis of air quality data allows the study of accidental events such as fires [18], the analysis of the effects of pollutant dispersion produced by industrial activities [19], and the implementation of forecasting models that can provide detailed information on air quality over time [20] and space. Knowing the behavior of pollutants, and in particular of airborne dust, is also useful for defining the levels of healthiness in closed environments, such as homes or workplaces. [21]. Unlike the mineral aerosol which is of natural origin, the PM10 produced in the urban environment comes from different anthropogenic sources. The major particulate matter sources in urban and extra-urban areas are represented by industrial plants, domestic heating systems, emissions due to vehicular traffic, tire breaking, agriculture, and livestock. In detail, the present study aims to evaluate the contribution of Saharan dust to the environmental pollution measured by the air quality monitoring stations in an urban context.

### The Case Study

The area investigated in this work is Avellino, a medium-sized city in Campania situated in a plain surrounded by mountains. The city of Avellino is one of the most polluted in central-southern Italy and seventh in the national ranking, with 78 days of overruns compared to the legal limit of 50 μg/m^3^ during 2020. The harsh winters and obsolete domestic heating technologies, added to the various industrial emissions and intense vehicular traffic, mean that the measured average levels of particulate matter PM10 and PM2.5 are very high. Moreover, the complex mountain topography does not facilitate the natural dispersion of pollutants. For all these reasons, the city is interested in monitoring air quality, and great attention is paid to the analysis of data from the regional monitoring stations.

## 2. Materials and Methods

### 2.1. Air Quality Sources

The control of the air quality parameters according to the regulatory provisions of Legislative Decree 155/2010 represents one of the main institutional activities of the Regional Agency for Environmental Protection of Campania region (ARPAC). The agency manages the monitoring network determined according to regional specifications for the identification and management of air quality monitoring stations, to assess the air quality and polluting emissions spread over the area. The configuration of the network in the Campania region (a southern Italian region) includes 36 fixed monitoring stations and 5 mobile laboratories (Figure 1).

The monitoring stations are located in sensitive areas, following the classification of the regional territory. There are also 10 additional fixed monitoring stations installed near the waste treatment plants (“STIR” network) that, although not part of the regional network, provide additional and support measures for the interpretation of the evolutionary phenomena of air quality on a regional basis.

The technology on which the measuring instruments of the ARPAC network are based comply with the technical standard UNI EN 12341: 2014. The measurement principle is gravimetry, and the reference method for the determination of PM10 particulate material is based on the collection of the PM10 fraction on a special filter and the subsequent determination of its mass by gravimetric methods in the laboratory, after conditioning the filter in controlled conditions of temperature (20 °C ± 1) and humidity (50 ± 5%).

In Avellino, the ARPA air quality monitoring network is composed of two monitoring stations, V° Circolo (VC) and Scuola Alighieri (SA), classified as suburban background and urban traffic, respectively. Both monitoring stations provide for PM10 and PM2.5 particulate matter measurements with a daily resolution using gravimetric devices and according to the legal specifications. The VC station also provides hourly particulate matter analysis by using laser scatter technology. Both stations also provide NO_2_ measurements, and SA adds measurements for CO and benzene, while VC adds only O_3_ data. The main characteristics of the stations considered in this work are summarized in Table 1. The SA station is located on one of the most congested roads of the city, while the VC station is located near a school that represents a sensitive point (see Figure 2). For this work, data for February 2021 were investigated.

In the city considered there is also an extra air quality monitoring station, part of the meteorological network managed by the non-profit organization MVOBSV. The sensor used is a PMSA003 Digital Laser Dust Sensor (MV) placed on a palace rooftop (18 m) in the city center (see Figure 2). The sensor is based on laser scattering technology with a time resolution of 10 min. The sensor uses the principle of laser dispersion, that is, by irradiating the suspension of particles in the air, it collects the scattered light at a certain angle and obtains a light dispersion curve that varies over time. The wavelength of the laser is 1100 μm. The particles of equivalent diameter and the number of particles with a different diameter per unit of volume can be calculated in real time by a microprocessor based on the MIE theory. In this case the sensor provides for PM10 and PM2.5 concentrations.

Since the ARPA data were compared with MVOBSV data that are based on the technology of laser scattering, a comparison with gravimetry was mandatory. The comparison between the two measurement techniques allowed for data validation.

Figure 3 reports the daily average PM10 and PM2.5 concentrations measured by the VC station with the two measurement techniques. From inspection of Figure 3, it is clear that the values measured with gravimetry and laser scattering are almost the same, with an average difference of 0.23% for PM10 and 0.5% for PM2.5. The good agreement of the data measured means that the laser scattering measurements are reliable for further analysis.

The MV station is placed on the roof 18 m from the ground, above the urban canyon, which acts as a mixing cell. This height should ensure that Saharan dust is not directly measured since it is mainly made up of PM10 and naturally tends to stagnate on the ground. The proposed approach consisted of the following fundamental steps:identification of the ARPA monitoring device with the same technological characteristics of the monitoring station to be compared, located at a distance, not exceeding 1000 m, and placed in the same urban context,validation of experimental data of basic pollution conditions through the analysis of comparison graphs,identification of Saharan events,comparison of the experimental daily values obtained by the measuring device with the ARPA values, and subsequently investigating in more depth the days with greater relevance using the hourly averages.

In this case study, the monitoring station analyzed was represented by MV and the ARPAC monitoring station used for the comparison was VC. The two stations considered (MV and VC) are both based on laser scattering technology and are located at a distance of 600 m in a context of medium urbanization.

### 2.2. Meteorological Data

Meteorological data are fundamental for pollution dispersion analysis. The data used in this study were provided by an automatic weather station (Davis Vantage Pro 2) co-located with an air quality sensor. This station is part of the meteorological network managed by the non-profit organization MVOBSV and includes sensors for the measurement of essential atmospheric parameters, i.e., air temperature, relative humidity, air pressure, rainfall, and wind speed and direction. The data are stored with a temporal resolution of 10 min and are fully available for the analyzed period.

## 3. Results

In the first analysis, the daily data provided by the MV station was compared with those of the station VC which is closest (Figure 4). Comparing the PM10 and PM2.5 trends of the stations showed a good agreement between the measurements. The measured data aligned around the central trend line, with small deviations, especially for PM2.5.

From the analysis of the PM10 and PM2.5 trends (Figure 5) it can be seen that for many days in February, both the recorded PM10 and PM2.5 exceeded the legal limit. In particular, the days exceeding the limit threshold of 50 μg/m^3^ for PM10 were 16 at the SA station and 17 at the VC station. For PM2.5, the daily limit of 25 μg/m^3^, similarly to PM10, was exceeded 16 times at the SA station and 17 times at the VC station. In February, the days when the pollution levels were low coincided with favorable weather conditions for dispersion. The three data trends have the same behavior, and differences can be attributed to the different positions in the urban context and, therefore, to the local conditions.

Subsequently, the trends of PM10, PM2.5, and their ratio were analyzed for each measuring station (Figure 6). An inspection of Figure 6 reveals that in the periods 5–7 and 23–28 of February, the difference between PM10 and PM2.5 concentrations was high for the SA and VC stations (Figure 6a,b); moreover, in such periods, the ratio between the particulate matter was low. This means that the contribution of PM10 to pollution was higher than PM2.5, and in particular, an external contribution represented by Saharan dust was added to the PM10 produced in the urban context. This external contribution was represented by the Saharan dust that in these days was dispersed over Italian territory.

The transported particulate is of desert and sandy origin, mainly composed of siliceous materials; unlike PM10 of urban origin which is characterized by the presence of carbon and metal-based particles. The analysis confirmed that the contribution of the mineral aerosol to the PM10 values can be very important, favoring the breaking of the legislative limit.

Looking in depth at the two periods selected, it is possible to analyze the hourly trends of pollutants (Figure 7 and Figure 8). In the first Saharan event, PM10 concentration reaches a peak of 180 μg/m^3^ measured at the VC station. As highlighted in Figure 7, the PM10/PM2.5 ratio was extremely variable at the VC station (Figure 7a), while it was slightly variable at the MV station (Figure 7b). Moreover, at the MV station there was no evidence of Saharan dust contributing to PM10 concentrations, which remained proportional to PM2.5.

Considering the second Saharan event, during 23–28 February and reported in Figure 8, the PM10 concentration reached an hourly peak of 260 μg/m^3^ measured by the VC station (Figure 8a). As in the previous case, the MV measuring station did not show any evidence of an external PM10 concentration contribution, in fact, the PM10/PM2.5 ratio was slightly variable and the pollutant trends were similar (Figure 8b). There was only a slight difference between PM10 and PM2.5 on days 26–27. The absence of evidence of a Saharan influence phenomenon at the MV station is probably due to the different height at which it is positioned compared to the ARPAC monitoring stations. The MV station is positioned on the roof of a building, while the two ARPAC stations are located near the street. The open-field exposure of the MV station does not allow the detection of the Saharan phenomenon, since the dust is more dispersed and diluted in the atmosphere at that height.

Further analysis included the daily pollution inspection, to compare the trends measured by the VC and MV stations. In particular, some of most interesting days were more deeply analyzed, and also including the meteorological parameters.

In Figure 9, the hourly measured concentration trends at the VC and MV stations appear to be in good agreement on most of the days analyzed. In detail, the measurements were also in agreement in their values and not only in their trend in 25% of cases.

On both days, at around 7 and 9 a.m. the pollutant levels increased. Particulate matter tends to settle on the ground due to gravity, both when airborne and as a result of rain. It remains on the asphalt if there is no movement of vehicles, such as during the night. When city activities start early in the morning and vehicles start moving on the roads, the particulate matter is dislodged from the road surface and released back into the atmosphere, increasing particulate concentrations. This phenomenon was more evident at the MV station than the VC station because it is located in the more congested city center.

Figure 10 shows the 500-hPa and sea level pressure fields for a typical scenario in which low pollutants concentrations were detected in the area of Avellino due to the simultaneous action of rain and winds. More specifically, the left panel of Figure 10 (Figure 10a) shows the synoptic scenario at 00:00 UTC on 11 February 2021, when the Italian peninsula was affected by a low-pressure area, well-structured at both mid and low tropospheric levels. This cyclonic system caused light rainfall in the study area (the AWS involved in this study registered a 24-h accumulated precipitation value of 1.5 mm), associated with moderate south-western winds. In the subsequent 12 h, the low-pressure area moved towards the Balkan peninsula (Figure 10b); this caused an improvement in weather conditions in Avellino. However, a moderate breeze from the west was still observed and, therefore, favorable pollutant dispersion renewed.

The Campania region was affected by two Saharan events in the periods 5–7 and 23–28 of February. Here, we provide a brief meteorological analysis of such events, and to better visualize the influence of Saharan dust on the Italian peninsula, synoptic maps were created.

Saharan event occurring on 5–7 February 2021.

On 5–7 February 2021, the European synoptic scenario resembled one of the most favorite patterns for the incoming Saharan air masses in the central Mediterranean area. The analysis of the 500-hPa geopotential height field revealed that, on 5 February (Figure 11a), the presence of a trough over Western Europe extended from the British Isles to the western sector of Morocco. This trough was the result of a relevant oscillation of the polar front that occurred in the East Atlantic between 2 and 4 February. The sea-level pressure distribution (also depicted in Figure 11a) showed a low-pressure area downstream of the trough axis, near the Gibraltar strait. A strong ridge, synonymous with stable weather conditions, instead modulated the atmospheric scenario in the Central and Eastern Mediterranean basins. In the subsequent 24 h, the trough moved eastwards (Figure 11b) and caused a drop of atmospheric pressure on the western side of the Italian peninsula. The consequence of this pressure pattern was a dry and warm meridional flow, which advected on the central Mediterranean area subtropical continental air masses coming from the Sahara region. In this respect, the 850-hPa equivalent potential temperature field presented in Figure 11c gives clear evidence of the warm tongue extending from the inland sectors of North Africa up to the Italian peninsula. On 7 February, the cyclonic system moved further to the east, causing a strengthening of warm air advection on southern Italy in the morning hours, and, consequently, a massive transport of Saharan dust. The meteorological conditions abruptly changed in the evening hours of 7 February, when the subtropical continental air was replaced by colder and moist air masses coming from the west.

Figure 11d offers a true-color image of the central and western Mediterranean basins, collected by The Moderate Resolution Imaging Spectroradiometer (MODIS) onboard NASA’s Terra satellite on 6 February 2021. This picture highlights the Saharan dust cloud rising from North Africa and crossing the Mediterranean Sea.

Saharan event occurring on 23–26 February 2021.

A second relevant Saharan event took place in the study area at the end of February 2021 (days 23rd–26th). The meteorological dynamics that forced the incoming of desert dust in the central Mediterranean area had some similarities with those involved in the previous event. To shed light on these dynamics, it is necessary to analyze the 500-hPa geopotential height field on 22 February at 12 UTC (Figure 12a), which reveals the presence of a cut-off low over the Alboran Sea, generated by a Rossby wave that affected the Iberian Peninsula and Morocco during 20 and 21 February. This upper-level circulation was associated with a low-level cyclonic area, located inland of Algeria, as clearly highlighted by the sea-level pressure field depicted in Figure 12a. On the Italian peninsula, stable weather conditions prevailed, due to the presence of a ridge. This synoptic scenario promoted an advection of warm sub-tropical continental air, which can be easily identified by means of the 850-hPa equivalent potential temperature field (see Figure 12b). The latter clearly shows the pattern followed by warm air, which reached central Europe after crossing the Sardinian Sea. On 23 February 2021, the ridge located over Italy strengthened (see Figure 12c), obstructing the natural eastward movement of the low-pressure area located over the western Mediterranean basins. This evolution led to the translation of the cyclonic system inland of Algeria and, therefore, to the end of the warm advection in the central Mediterranean area. The previously advected sub-tropical air (rich in Saharan dust), following a clockwise motion imposed by the anticyclone, spread out over the Italian peninsula and Balkan regions. Owing to the persistence of the ridge in the subsequent days, the Saharan dust cloud continued to flow around Italy although it had been inevitably subjected to a gradual dilution process.

Satellite evidence of Saharan dust affecting central and northern Italy is provided by the image in Figure 12d, acquired by MODIS on 23 February 2021.

## 4. Discussion

The analysis of Saharan dust contribution to PM10 in urban areas is defined by the high levels of PM10 measured not followed by a simultaneous increase in PM2.5 concentrations. By applying to this case study the method proposed by Escudero et al. [22], it was possible to define the contribution of PM10 measured on the days in which the Saharan events occurred, using the 30th percentile of the monthly concentrations. By applying this procedure to the background VC station, we obtained the PM10 value in the absence of mineral aerosol. Comparing the values obtained after the application of the procedure suggested by Escudero et al., PM10 values similar to those measured at the MV station were obtained. As an example for the VC station, a daily value of 60 μg/m^3^ was measured, and the monthly 30th percentile value was 27 for this month. Therefore, the net dust contribution at this point, for this day, was 33 μg/m^3^. For the same day considered, the MV station recorded a PM10 daily value of 31 μg/m^3^. Comparing this last value with the previous one obtained without considering the mineral dust contribution to the PM10, it is clear that they are similar. This confirms that the Saharan dust phenomenon was not measured from the station located at a height of 15/20 m, as suggested by the method proposed in this study. With this approach it is possible to also evaluate the exceedance in PM10 concentrations due to the mineral dust to the urban background.

This type of approach can be used in any city with orographic characteristics, urbanization, industrial supplies, population density, type of heating, and degree of traffic similar to the city in the case study. The peculiarity of being in a basin, surrounded by mountains is a frequent feature of Italian and European cities, which makes the case study adaptable to many real situations, such as the Po valley in Lombardy (Italy).

## 5. Conclusions

The occurrence of natural events such as the transport of Saharan dust can lead to a significant increase in the PM10 concentrations measured in urban areas. The correct analysis of the air quality data, including the evaluation of the influence of these natural phenomena, is necessary to define the real levels of urban pollution. The Saharan phenomena bring the levels of PM10 to very high values, as in this case study to 260 μg/m^3^, which if interpreted incorrectly can generate unnecessary alarmism in the population and institutions. In fact, due to their nature, the events that transport Saharan dust are completely unpredictable and therefore unmanageable by man. What can be done to lower pollution levels is the reduction in the hours of ignition of domestic heating and the use of eco-friendly fuels, a significant reduction in traffic emissions, and the control of emissions from agriculture. Surely, in a territory that is based on an economy including agriculture, the elimination of phenomena such as weed burning would have a positive influence on the reduction of pollution. Furthermore, it must be considered that the territory of this case study, as previously mentioned, has an orography that is unfavorable to pollutant dispersion, therefore it is necessary to reduce the levels of pollution, which tend to rise due to the low natural dispersion. The implementation within the urban context of an air quality monitoring network is of fundamental importance for the definition of the city pollution levels. The regional networks of environmental protection agencies (ARPA) must be integrated with additional monitoring stations to obtain a greater spatial resolution of the data. Furthermore, the implementation of further measurement stations based on IoT technology allows real-time knowledge of the levels of airborne pollutants. The spatial and temporal detail of the measured data is the starting point for having a clear vision of the pollution phenomena, which can be analyzed in detail, also highlighting natural phenomena such as Saharan dust. From this perspective, the MV air quality monitoring device of the Mt.Vergine Observatory is placed such that, with the latest generation technology, it provides data to support traditional technologies.

## Figures and Tables

**Figure 1 ijerph-18-06100-f001:**
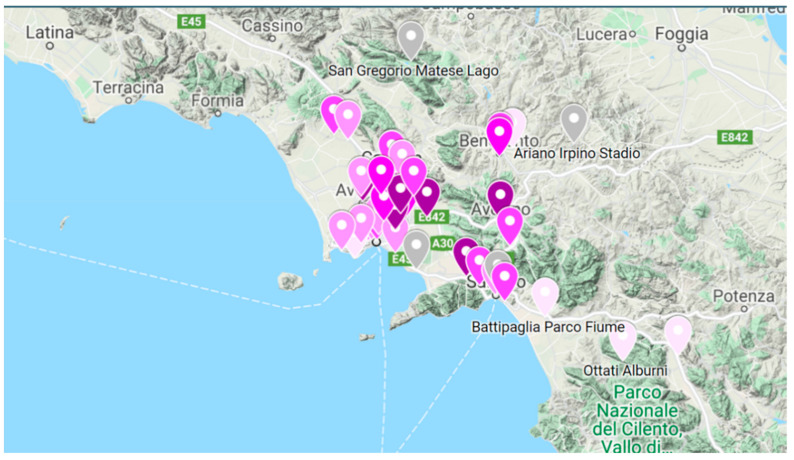
ARPA air quality monitoring network located in the Campania region.

**Figure 2 ijerph-18-06100-f002:**
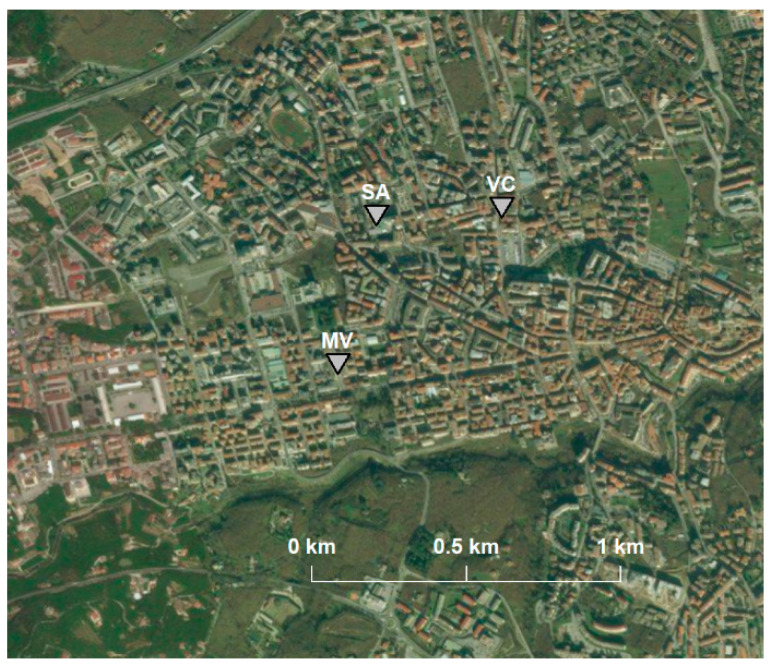
Map of measuring stations located in Avellino.

**Figure 3 ijerph-18-06100-f003:**
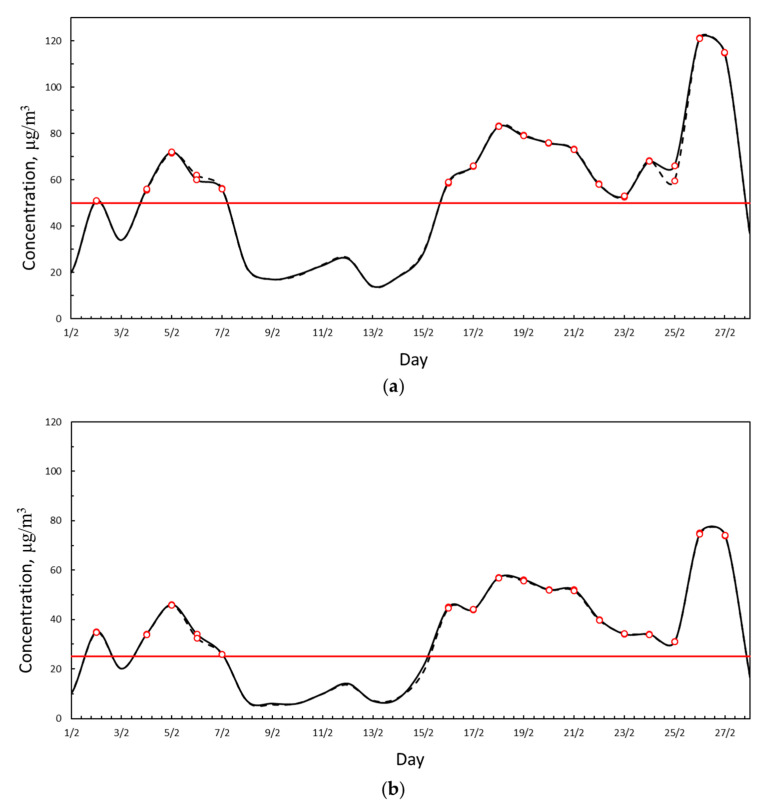
(**a**) PM10 daily average concentration measured at the VC station with gravimetry (solid black line) and with laser scattering (dashed black line). Red line represents the legal limit concentration according to D.Lgs. 155/2010. Red dots represent exceedance of the legal limit. (**b**) PM2.5 daily average concentration measured at the VC station with gravimetry (solid black line) and with laser scattering (dashed black line). Red line represents the legal limit concentration according to D.Lgs. 155/2010. Red dots represent exceedance of the legal limit.

**Figure 4 ijerph-18-06100-f004:**
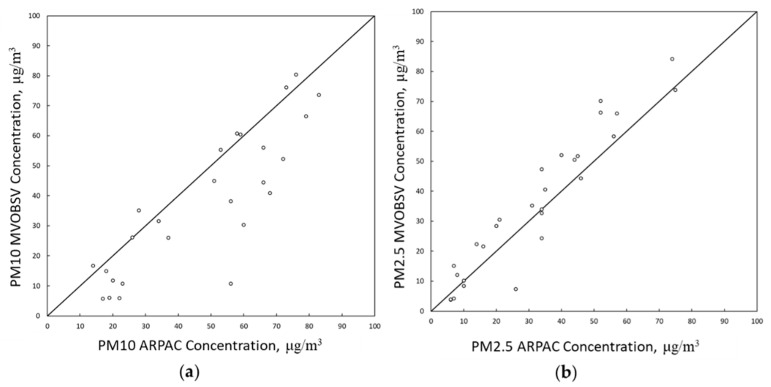
(**a**) Comparison of PM10 daily average concentrations measured at the MV and VC stations. (**b**) Comparison of PM2.5 daily average concentrations measured at the MV and VC stations.

**Figure 5 ijerph-18-06100-f005:**
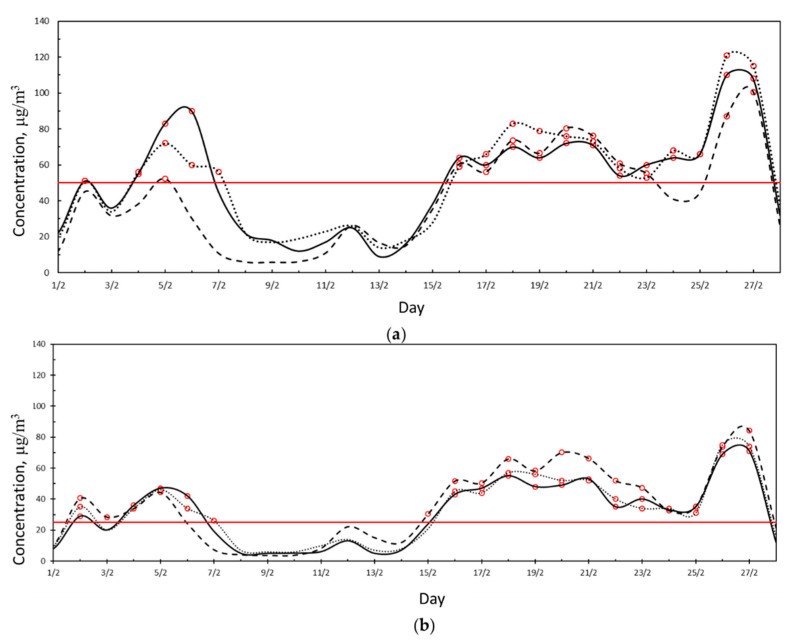
(**a**) PM10 daily average concentration measured at the SA station (solid black line), VC station (dotted black line), and MV station (dashed black line). Red line represents the legal limit concentration according to D.Lgs. 155/2010. Red dots represent exceedance with respect to the legal limit. (**b**) PM2.5 daily average concentration measured at the SA station (solid black line), VC station (dotted black line), and MV station (dashed black line). Red line represents the legal limit concentration according to D.Lgs. 155/2010. Red dots represent exceedance of the legal limit.

**Figure 6 ijerph-18-06100-f006:**
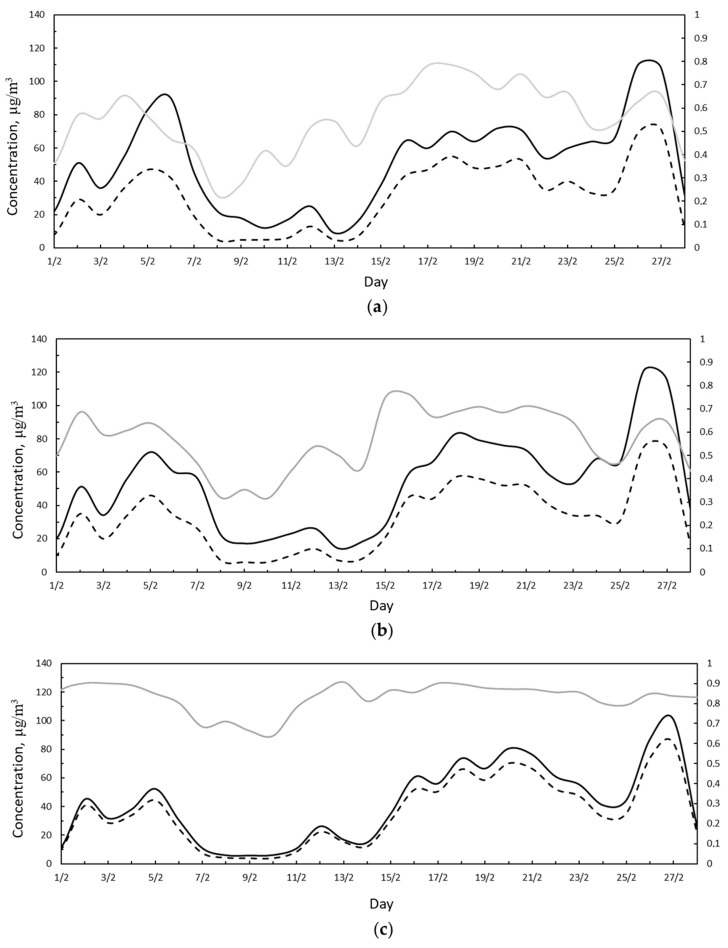
(**a**) Daily PM10 (black solid line), PM2.5 (black dashed line), and PM2.5/PM10 concentrations (grey solid line) measured during February 2021 at the SA station. (**b**) Daily PM10 (black solid line), PM2.5 (black dashed line), and PM2.5/PM10 concentrations (grey solid line) measured during February 2021 at the VC station. (**c**) Daily PM10 (black solid line), PM2.5 (black dashed line), and PM2.5/PM10 concentrations (grey solid line) measured during February 2021 at the MV station.

**Figure 7 ijerph-18-06100-f007:**
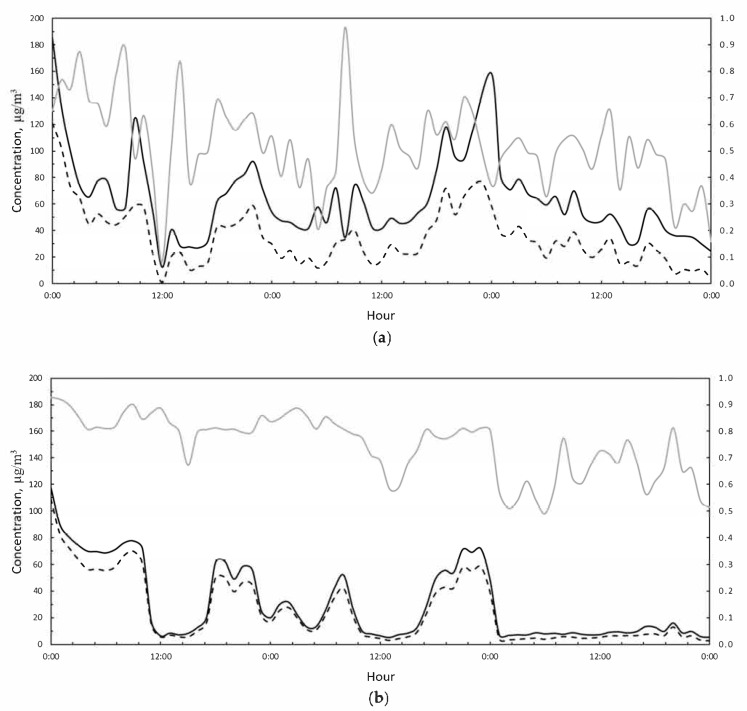
(**a**) PM10 (black solid line), PM2.5 (black dashed line), and PM2.5/PM10 hourly concentrations (grey solid line) measured during the period 5–7 February 2021 at the VC station. (**b**) PM10 (black solid line), PM2.5 (black dashed line), and PM2.5/PM10 hourly concentrations (grey solid line) measured during the period 5–7 February 2021 at the MV station.

**Figure 8 ijerph-18-06100-f008:**
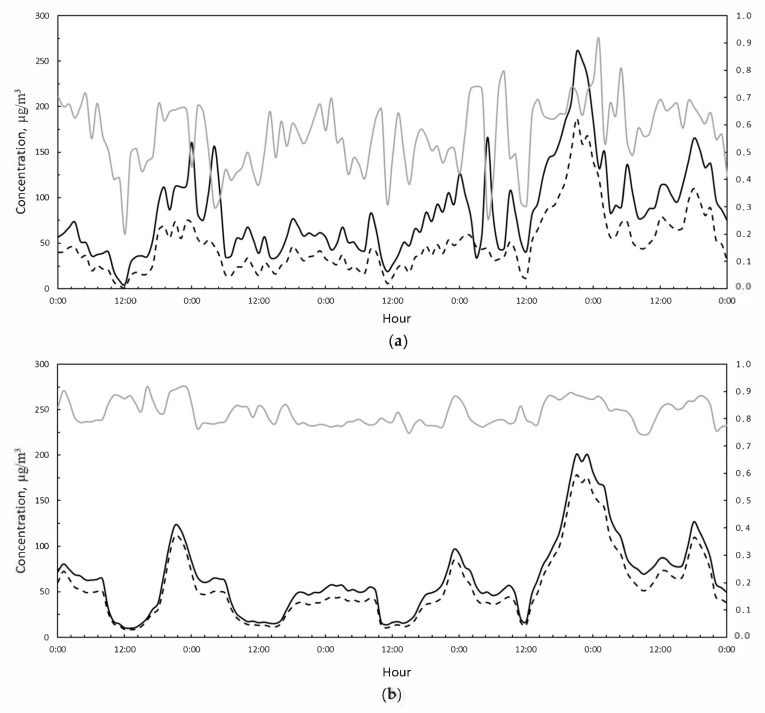
(**a**) PM10 (black solid line), PM2.5 (black dashed line), and PM2.5/PM10 hourly concentrations (grey solid line) measured during the period 23–28 February 2021 at the VC station. (**b**) PM10 (black solid line), PM2.5 (black dashed line), and PM2.5/PM10 hourly concentrations (grey solid line) measured during the period 23–28 February 2021 at the MV station.

**Figure 9 ijerph-18-06100-f009:**
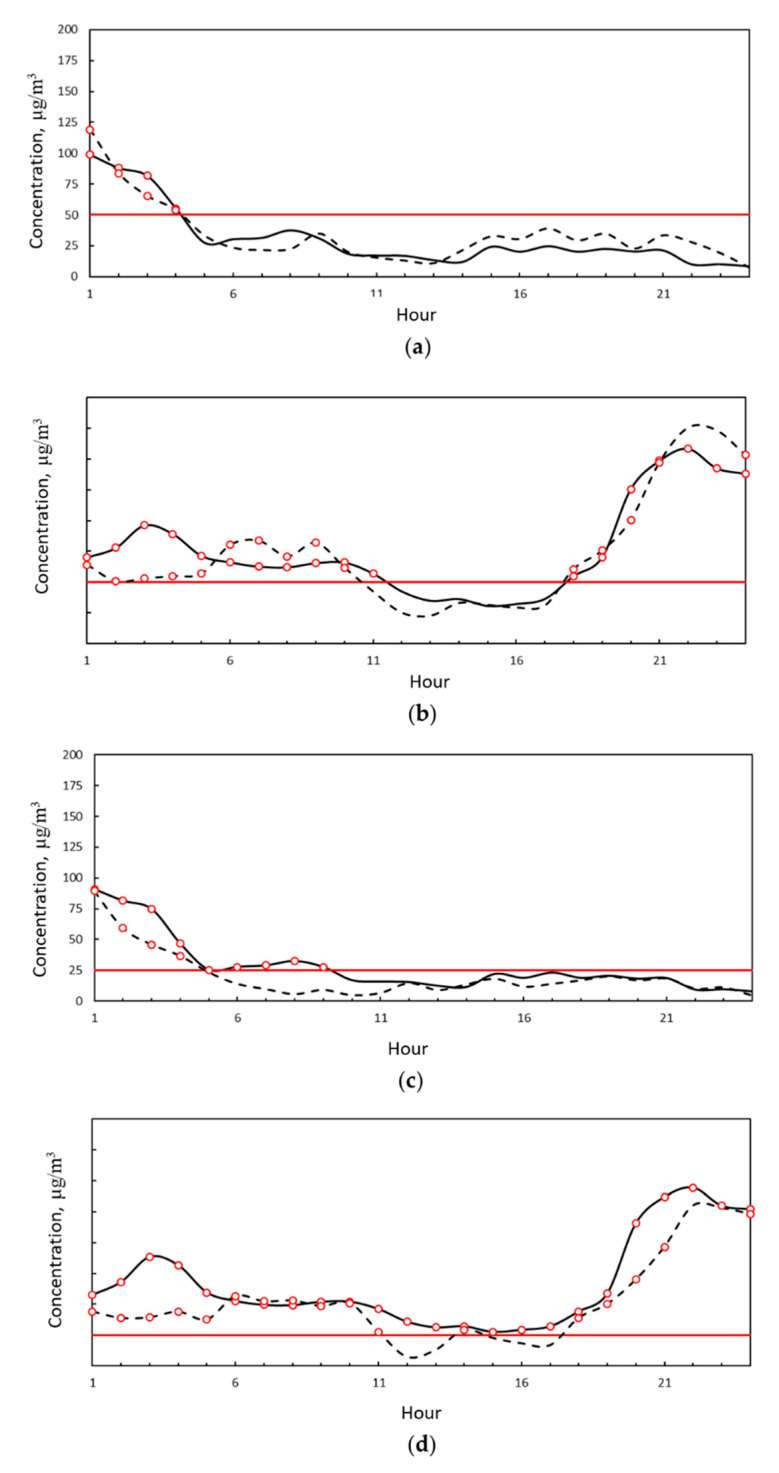
(**a**) PM10 hourly average concentration measured at the VC station (solid black line) and MV station (dashed black line) 3 February 2021. Red line represents the legal limit concentration according to D.Lgs. 155/2010. Red dots represent exceedance of the legal limit. (**b**) PM10 hourly average concentration measured at the VC station (solid black line) and MV station (dashed black line) 21 February 2021. Red line represents the legal limit concentration according to D.Lgs. 155/2010. Red dots represent exceedance of the legal limit. (**c**) PM10 hourly average concentration measured at the VC station (solid black line) and MV station (dashed black line) 3 February 2021. Red line represents the legal limit concentration according to D.Lgs. 155/2010. Red dots represent exceedance of the legal limit. (**d**) PM2.5 hourly average concentration measured at the VC station (solid black line) and MV station (dashed black line) 21 February 2021. Red line represents the legal limit concentration according to D.Lgs. 155/2010. Red dots represent exceedance of the legal limit.

**Figure 10 ijerph-18-06100-f010:**
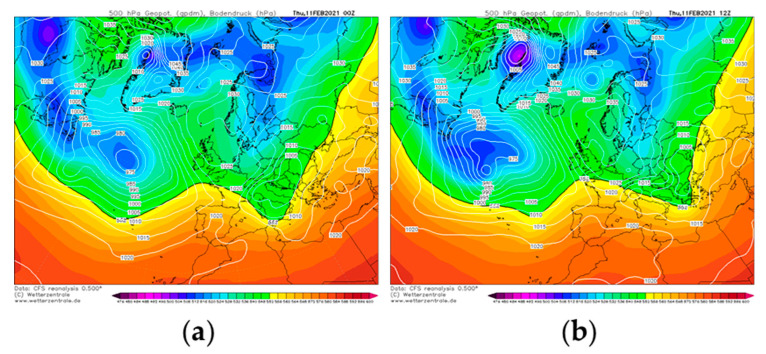
(**a**) Geopotential height at 500 hPa (shaded color) and sea level pressure (white solid line) fields at 00:00 UTC on 11 February 2021. (**b**) Geopotential height at 500 hPa (shaded color) and sea level pressure (white solid line) fields at 12:00 UTC on 11 February 2021 and 12:00 UTC. Both fields were retrieved from Climate Forecast System Renalysis of the National Centers for Environmental Prediction.

**Figure 11 ijerph-18-06100-f011:**
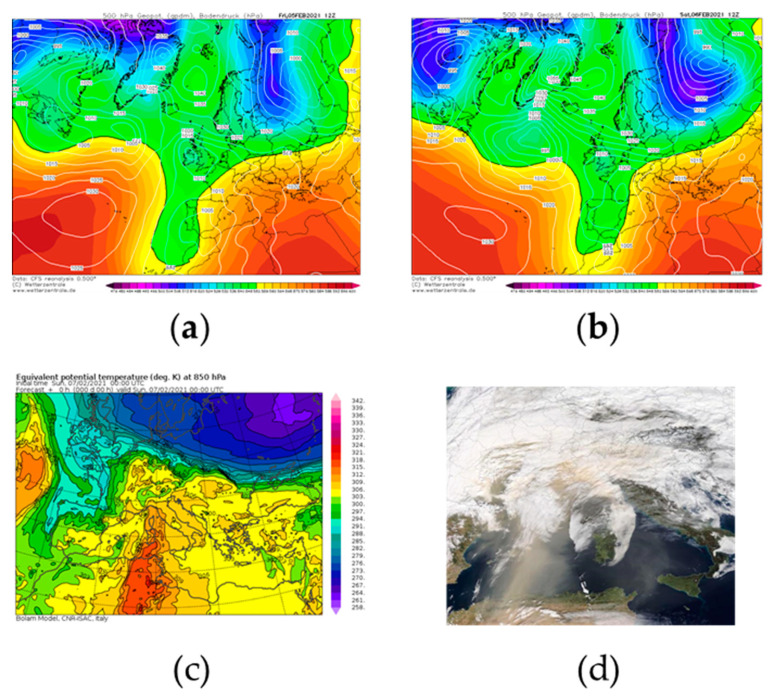
(**a**) Geopotential height at 500 hPa (shaded color) and sea level pressure (white solid line) fields at 12:00 UTC of 5 February 2021. (**b**) Geopotential height at 500 hPa (shaded color) and sea level pressure (white solid line) fields at 12:00 UTC 6 February 2021. Both fields were retrieved from Climate Forecast System Renalysis of National Centers for Environmental Prediction. (**c**) 850 mb equivalent potential temperature (shaded color) at 00:00 UTC 7 February 2021 from the BOLAM model of the Institute of Atmospheric Science and Climate of the Italian National Research Council. (**d**) Image of the Saharan dust event acquired on 6 February 2021 by the Moderate Resolution Imaging Spectroradiometer (MODIS) onboard NASA’s Terra satellite. Images in panels (**a**) and (**b**) are courtesy of http://www.wetterzentrale.de, (accessed on 23 February 2021) whereas the image in (**c**) was retrieved from https://www.isac.cnr.it/dinamica/projects/forecasts/index.html. (accessed on 23 February 2021) The picture in panel (**d**) is courtesy of the NASA Worldview Snapshots application (https://wvs.earthdata.nasa.gov) (accessed on 23 February 2021).

**Figure 12 ijerph-18-06100-f012:**
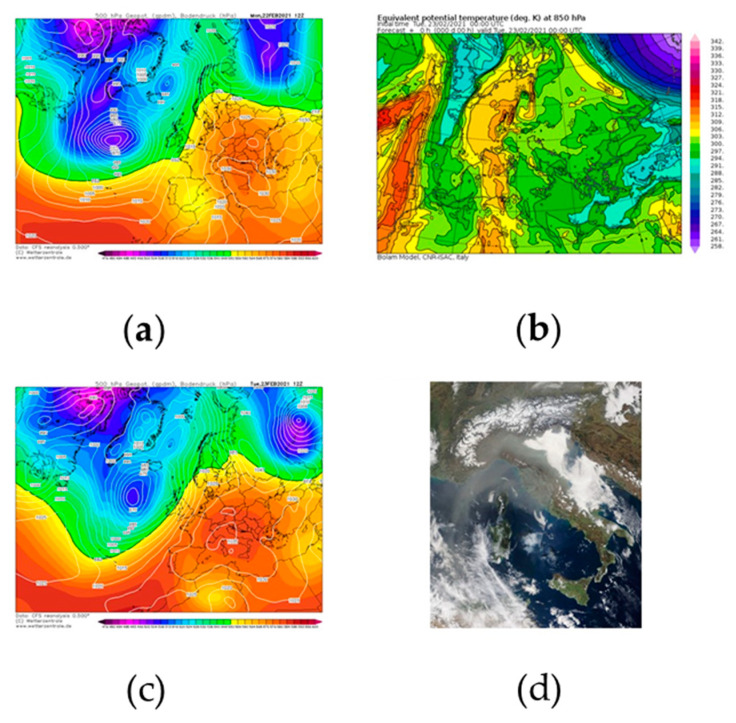
(**a**) Geopotential height at 500 hPa (shaded color) and sea level pressure (white solid line) fields at 12:00 UTC of 22 February 2021. Fields were retrieved from Climate Forecast System Renalysis of National Centers for Environmental Prediction. (**b**) 850 mb equivalent potential temperature (shaded color) at 00:00 UTC of 23 February 2021 from the BOLAM model of the Institute of Atmospheric Science and Climate of the Italian National Research Council. (**c**) Geopotential height at 500 hPa (shaded color) and sea level pressure (white solid line) fields at 12:00 UTC of 23 February 2021. Fields were retrieved from Climate Forecast System Renalysis of National Centers for Environmental Prediction. (**d**) Image of the Saharan dust event acquired on 23 February 2021 by the Moderate Resolution Imaging Spectroradiometer (MODIS) onboard NASA’s Terra satellite. Images in panels (**a**) and (**b**) are courtesy of http://www.wetterzentrale.de (accessed on 23 February 2021, whereas the image in (**c**) was retrieved from https://www.isac.cnr.it/dinamica/projects/forecasts/index.html (accessed on 23 February 2021). The picture in panel (**d**) is courtesy of the NASA Worldview Snapshots application (https://wvs.earthdata.nasa.gov) (accessed on 23 February 2021).

**Table 1 ijerph-18-06100-t001:** Characteristics of the monitoring stations.

Station	Acronym	Type	Measurements Technique	
**V° Circolo**	VC	Suburban background	Gravimetric Laser scattering	ARPAC
**Scuola Alighieri**	SA	Urban traffic	Gravimetric	ARPAC
**Mt. Vergine Observatory**	MV		Laser scattering	MVOBSV

## Data Availability

Data are available at https://www.arpacampania.it/aria.

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
