# Peer review of "An Innovative Approach to Determining the Contribution of Saharan Dust to Pollution"

_ijerph, 2021, doi:10.3390/ijerph18116100_

Round 1

Reviewer 1 Report

The purpose of this paper was to analyze the air quality data of a regional monitoring network (located in Italy) in integration with a monitoring station based on IoT technology to focus on the major problems in pollution levels over a city in Italy during February 2021. This paper was reviewed carefully and it has been judged that it should be rejected. I do not recommend the publication of this work based on the major issues that are listed below:

  1. In the title of the paper, the authors mentioned that they have an innovative approach in this study, while not only the Materials and Methods section does not represent a new method, but also there is a major lack of sufficient and explicit explanation about the methodology. The authors should describe the measurement techniques clearly and in detail. How the V Circolo (VC), Scuola Alighieri (SA), and MT.Vergine Observatory (MV) stations (which the results are based on their measurements) measure the particle (in detail)? Using laser scattering technology or gravimetric, but how? at what wavelength range? what angle? It is just mentioned that they work based on the Gravimetric and Laser Scattering but there is a serious lack of sufficient explanation in the methodology. Defining some theories could not convince me at all that they have an innovation in the methodology.

- (lines 155-159: Microprocessor using based on the Mie theory of the algorithm, get the particle equivalent particle size and unit volume of particles of different particle size! The measuring device provides for the PM10 and PM2.5 concentrations! What is the meaning of these sentences exactly??)

  1. Where is the Discussion section of this paper? Section 3 represents the Results and suddenly it goes to section 5 Conclusion!! I am wondering that how is it possible that the authors did not compare their findings with any other similar studies in that past? or similar methodology? or even close studies at the same location (at least in Europe)? how come that they did not discuss their findings in detail? The Discussion section is an essential section of a paper which in this paper is not provided!

  1. The authors just simply concluded, without even mentioning one reference (please note that the last reference mentioned in this paper is number 20, which is related to the Introduction section). Through whole the result section the authors provided some reasons for their findings, how the readers can accept these reasons? Are there any other studies with similar findings or opposite ones that readers can refer to them?

  • lines 319-334 the authors simply and without even mentioning one reference provided some reasons for their findings. how do you convince the readers that the reason for the findings is scientific?

Author Response

Dear reviewer, we have implemented your suggestions, the work looks much improved. Thank you very much

Reviewer 2 Report

The submitted paper is good on the whole, however a revision is needed prior to publication, in particular the following three points could help improving the present version of the paper:

1) The structure of the paper might be changed, in the sense that the Saharan dust phenomena should be presented first, and then the Avellino case study should be discussed as application.

2) It should be clearer why Avellino is the case study, and in which other places this study might be extrapolated?

3) To further highlight integrated approaches to the problem of dust, there is a recent special issue on dust and the environment that might be consulted (Doronzo, D.M.; Al-Dousari, A. Preface to Dust Events in the Environment. Sustainability 2019, 11, 628. https://doi.org/10.3390/su11030628).

Best regards

Author Response

(The authors gave the same response as above.)

Reviewer 3 Report

Overall this manuscript provides a timely report on the air quality in Avellino. And the effect of significant events, like the Saharan event, on air quality is also discussed. The conclusions are reasonable. Some points might be addressed to further improve its quality:

  1. Introduce the Saharan event in the Introduction part, especially the activities that might cause air pollutions.
  2. A detailed comparison of MVOBSV and ARPAC should be provided.
  3. No monitoring stations are shown in Figure 1. Need to fix it.
  4. In Figure 4, use different symbols to show the data from MVOBSV and ARPAC.
  5. The geopotential height at 500 hPa (shaded col-our) and sea level pressure for a low-pollution day, like Feb. 11th, should be provided for a better comparison.
  6. After section 3, there is no section 4, but it directly becomes section 5.

Author Response

(The authors gave the same response as above.)

Round 2

Reviewer 1 Report

The manuscript was thoroughly checked for the major problems. The authors have added a proper explanation about the methodology with almost sufficient detail about the measurement principles and study area, however, the innovation in the measurement method is still questionable. A discussion section (Section 4. Discussion) has been added to the manuscript, although only one reference is not acceptable for the comparison with the close methodologies or the same study area but the discussion section can be admissible. The authors have deleted the vague sentences with serious language problems. In general, the current manuscript can be accepted after checking the comments as below:

==> It is expected that the authors compare their methodology and results with more previous studies. Even though the authors declare that they have novelty in their methodology, however, there should be previous studies with the close method or at the same study area or at least different method but similar/different findings around the study area.

==> Minor English grammar and spell check is needed, for example, line 283: In the city considered there also an extra air quality monitoring station…)

==> Figure 3 (a) and (b): Please increase font size for number of days and concentrations.

==> Same for Figure 4 (a) and (b), and Figure 5: Please increase the font size for the numbers.

==> Line 228: The technology on which the measuring instruments of the ARPAC network are 227 based refer to the technical standard UNI EN 12341: 2014 (It is preferred that a brief explanation be added for the readers who are less familiar with this standard; for example, EN 12341 (Standard method for determining the PM10 or PM2.5 mass concentrations of suspended particulate matter…).

Reviewer 3 Report

This version contains the necessary information and should be ready to go.